# EasyBalance: Cross-Layer Load Balancing in Distributed MoE Inference

Yize Wu [1 2]   Ke Gao [1]   Ling Li [1 2]   Yanjun Wu [1]

## Abstract

Load Balancing has emerged as a critical problem in expert-parallel distributed inference of Mixture-of-Experts (MoE) models. As routing distributions are typically skewed across experts, devices hosting lighter-loaded experts must idle to wait for the heaviest during expert computing, leading to inefficiency. Existing load-balancing approaches primarily rely on expert replication or migration within each layer, which introduce additional overhead and limit their flexibility and scalability. To address this problem, we propose EasyBalance, a **cross-layer** load balancing strategy that requires no modifications to the expert-device mapping, enabling instant adaptability and incurring essentially no additional overhead. Our key insights are that (1) experts from other layers can be viewed as naturally redundant, and (2) expert workloads of multiple layers can be jointly executed. Based on these observations, EasyBalance greedily schedules a subset of cross-layer workloads at each MoE-computation stage to run, while deferring the remaining workloads for future balancing opportunities. Extensive experiments across different models, tasks, and parallelism configurations demonstrate that EasyBalance consistently accelerates distributed MoE inference, reducing GPU idling by mostly over 40%.

## 1. Introduction

The rapidly growing parameter sizes of transformer-based models (Vaswani et al., 2017) present substantial computational challenges (Hoffmann et al., 2022). Mixture-of-Expert (MoE) architecture (Jacobs et al., 1991) offers a solution by replacing dense feed-forward layers with multiple smaller expert networks, where each token activates only a sparse subset of them. This sparse activation mechanism significantly reduces per-token computation cost (Fedus et al., 2022), while enabling models with substantially larger overall parameter counts.

In distributed MoE inference, experts are typically dispatched across multiple devices to parallelize their computation workloads, known as expert parallelism. In this setting, when tokens are routed to experts residing on remote devices, the tensors of their hidden representations are first dispatched to the corresponding devices via an all-to-all communication. Then, after expert computation, a second all-to-all gathers the results for aggregation (Huang et al., 2024; Hwang et al., 2023).

While expert parallelism can substantially accelerate MoE inference, load imbalance has emerged as a critical performance bottleneck (Liu et al., 2024). Despite the use of load-balancing auxiliary loss during MoE training, inference-time token routing often remains uneven across experts, leading to skewed computational workloads among devices. All the devices must wait for the most heavily loaded one to finish its computation, as the operations of result gathering and aggregation require synchronization. This results in severe resource under-utilization and system inefficiency (Liu et al., 2025b).

Existing methods for mitigating load balancing primarily rely on expert replication or migration within each expert layer(Li et al., 2023; DeepSeek AI, 2025; Doucet et al., 2025). By modifying the expert–device mapping of MoE models in the system, these methods attempt to redistribute workloads more evenly across devices. Despite their effectiveness, such approaches suffer from two inherent limitations. First, they lack flexibility when serving newly assigned tasks, as expert routing patterns can vary substantially across inputs and applications (see Figure 7). Second, expert replication incurs considerable memory overhead, while expert migration introduces additional communication costs, largely limiting their scalability (demonstrated in Section C.2).

To overcome these limitations, we propose EasyBalance, a novel cross-layer load balancing strategy. Unlike prior approaches, EasyBalance requires no changes to the expert–device mapping throughout the inference process,

---

[1]Intelligent Software Research Center, Institute of Software, CAS, Beijing, China [2]University of Chinese Academy of Sciences, Beijing, China. Correspondence to: Yanjun Wu <yanjun@iscas.ac.cn>.

*Proceedings of the 43rd International Conference on Machine Learning*, Seoul, South Korea. PMLR 306, 2026. Copyright 2026 by the author(s).

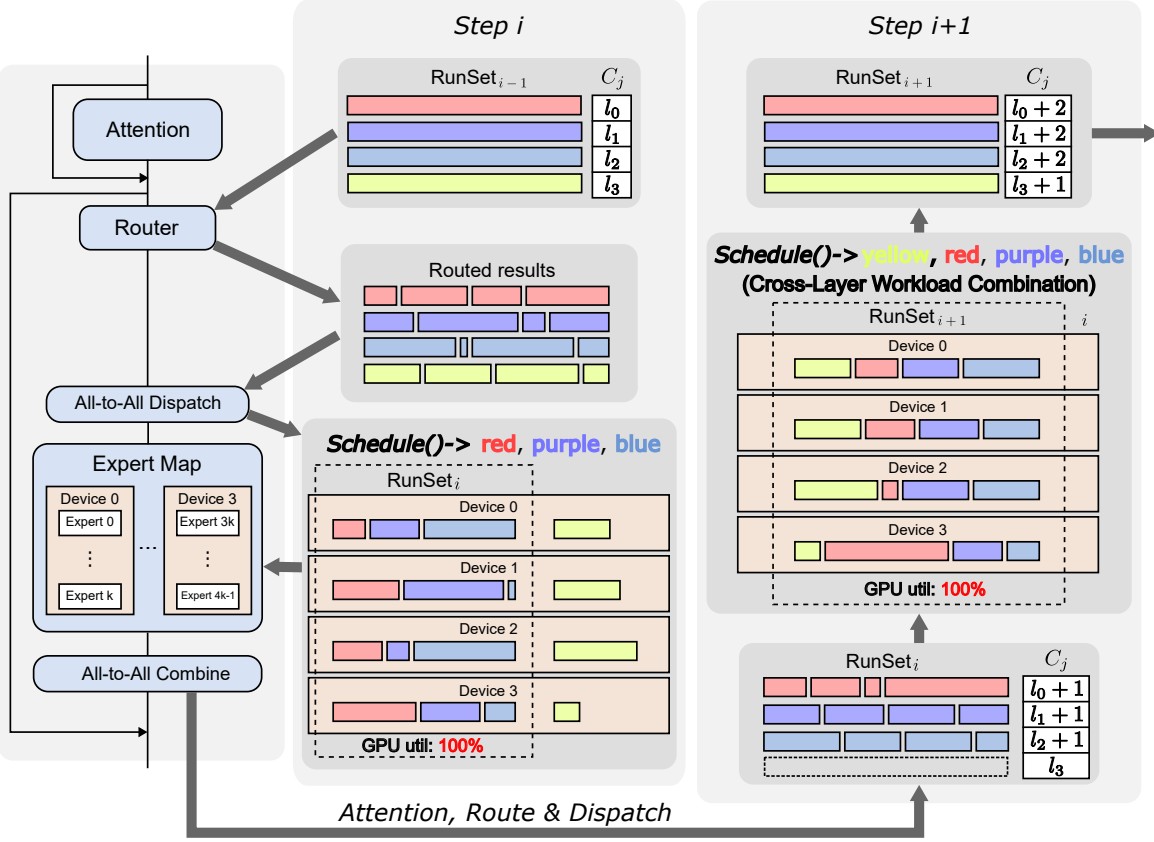

*Figure 1.* Demonstration of EasyBalance. Expert computation workloads of different micro-batches can be seletively executed and combined, achieving improved GPU utilization. $C_i$ indicates the current layer of each micro-batch.

thereby providing instant task adaptability with essentially no additional memory or communication overhead. Easy-Balance is motivated by two key observations.

**Cross-Layer Expert Redundancy.** Expert replication (DeepSeek AI, 2025) expands load-balancing opportunities by creating redundant experts, but at the cost of increased memory consumption. However, from a novel cross-layer perspective, we observe that redundancy already exists in an inference system: for a given expert layer, experts in other layers can be regarded as naturally "redundant"', as they already reside in device memory (just like the replication). These experts can therefore be leveraged to balance workloads without introducing any additional operations, yet this potential remains largely unexplored.

**Cross-Layer Workload Combination.** While model inference must follow a strict layer-wise sequential order, different micro-batches can progress through multiple layers simultaneously. This allows expert workloads from multiple layers to be executed concurrently. As shown in Figure 1, the sequences can run on different layers ($l_i$ and $l_j$ can be different for $i \neq j$, $i, j \in \{0, 1, 2, 3\}$).

Furthermore, in terms of performances, such workload

combination is safe in the worst case and beneficial in all other scenarios: for two workload distributions $w_1$ and $w_2$, MoE computation is bottlenecked by the heaviest dispatch, i.e., $\max(w_1)$ and $\max(w_2)$, and we have $\max(w_1 + w_2) \leq \max(w_1) + \max(w_2)$. The worst case occurs when workloads skew on the same device, while the probability decreases with the number of devices growing. Therefore, cross-layer workload combination is typically beneficial practically, especially in distributed scenarios (see detailed explanation in Section 4.1, and empirical evidences in Table 2).

Based on these observations, EasyBalance performs cross-layer scheduling over micro-batches to mitigate load imbalance within the whole inference process. At each expert computation stage, it selectively schedules a subset of workloads for execution, while deferring others to wait for potential future balancing opportunities. The scheduling algorithm greedily selects the workload combination maximizing GPU utilization, while other variant algorithms based on alternative scheduling heuristics have also been shown to be effective. Experimental results across models and tasks demonstrate that EasyBalance can consistently accelerate the inference process by mitigating load imbalance,

with GPU under-utilization reduced by mostly over 40%.

## 2. Preliminary

### 2.1. Mixture-of-Expert Architecture

A transformer layer typically consists of two main modules: self-attention and multi-layer perceptron (MLP). Given the input hidden state $h_l$ of layer $l$, the output of each module is aggregated with residual connections (He et al., 2016) as:

$$h'_l = h_l + \text{Attention}(h_l)$$

$$h_{l+1} = h'_l + \text{MLP}(h'_l)$$

The Mixture-of-Expert (MoE) architecture replaces the dense MLP with a set of experts, which are sparsely activated through a routing mechanism. For each token, the router selects the top-$k$ experts according to the gating function $G$:

$$W_l, I_l = \text{topk}(G(h_l)).$$

Each selected expert processes the token independently, and their outputs are aggregated using a weighted sum:

$$MLP(h_i) = \sum_{i \in I_l} W_{l,i} * \text{expert}_i(h_l), \qquad (1)$$

where $W_{l,i}$ denotes the routing weight for expert $i$. The aggregated result is then passed to the next layer as the output of the MoE block.

### 2.2. Expert Parallelism

In distributed MoE inference, expert parallelism is commonly adopted to dispatch expert computation across multiple devices. Each expert is placed onto one or more specific devices according to a expert–device mapping (typically an even distribution). Each device hosts a subset of experts and processes tokens routed to them.

Tokens may be assigned to experts that reside on remote devices. To solve this, an all-to-all communication is required to dispatch token representations to the corresponding devices. After computation, the expert outputs are communicated back to the original devices, where the weighted aggregation is performed.

### 2.3. Load Balancing

Although MoE training typically incorporates an auxiliary load-balancing loss to encourage uniform token routing, the routing distribution at inference time often remains skewed, leading to uneven workload distribution across devices under expert parallelism. As the gathering communication and weighted aggregation are both synchronous operations,

| Model | TTFT | |
|---|---|---|
| | w/o micro-batch | w/ micro-batch (4 splits) |
| Q3-30B | 45.47 | 41.61 |
| M-16B | 21.05 | 20.29 |

*Table 1.* Time to First Token (TTFT) of Qwen3-30B (Q3-30B) and Moonlight-16B (M-16B) on 2wikimqa. The sequence length and batch size are 4K and 64.

devices with lighter workloads must idle while waiting for the most heavily loaded one to finish, introducing under-utilization and therefore suboptimal performances.

Formally, let the expert computation workload at layer $l$ across $D$ devices as $w^{(l)} = (w_1^{(l)}, \cdots, w_D^{(l)})$. Distributed MoE computation is bottlenecked by the slowest device, so the effective workload is:

$$\hat{w} = \sum_{l=1}^{L_{max}} \max(w^{(l)}), \qquad (2)$$

which is directly proportional to execution latency. Moreover, the averaged computation-resource utilization across all the layers is defined as:

$$u = \frac{\sum_{l=1}^{L_{max}} \sum_{i=1}^{D} w_i^{(l)}}{\sum_{l=1}^{L_{max}} \hat{w} \times D}, \qquad (3)$$

which quantifies hardware efficiency by measuring the ratio of used resources to total available capacity.

### 2.4. Micro-Batching

Micro-batching is a widely used optimization method in large-scale inference scenarios. It splits a large batch of input sequences into multiple micro-batches and pipelines their execution (Kwon et al., 2023; SGLang Team, 2024). This allows communication for one micro-batch to be overlapped with computation for others, effectively hiding communication latency (Wang et al., 2025). It also enables a larger batch to fit in the memory by piecewise computation, and can also benefit KV cache transfer in prefilling-decoding disaggregation (Yan et al., 2025).

Empirical results in Table 1 demonstrate that the communication overheads can be significantly amortized using micro-batching. EasyBalance builds upon micro-batching to further enable cross-layer workload balancing during expert computation.

## 3. Problem

Existing approaches address load imbalance through expert replication and/or migration within each expert layer, to achieve fine-grained redistribution of workloads across

| Task | $N$ | |
|---|---|---|
| | 2 | 3 |
| 2wikimqa | 13/188 | 0/184 |
| trec | 19/188 | 4/184 |
| repobench-p | 43/188 | 4/184 |
| all | 518/3948=0.131 | 25/3864=0.006 |

*Table 2.* Number of $N$ consecutive layers that have identical workload distribution skew. The tested model is Qwen3-30B-A3B-Instruct. "all" stands for the statistics from all tasks in LongBench.

devices. Experts replication allow tokens to be selectively routed to less-loaded GPUs, and expert migration alleviates workload hot-spots by relocating heavily loaded experts to other GPUs. These techniques typically require routing-distribution statistics for modifying the expert mapping.

Despite their effectiveness, current solutions suffer from two inherent limitations that restrict their applicability in broader inference scenarios:

**Flexibility**. As shown in Figure 7, routing patterns of MoE layers are highly task-dependent. Therefore, a successful expert mapping for a specific task may become suboptimal— or even harmful—for inputs from other tasks. The reliance on task- or distribution-specific information fundamentally limits the flexibility of existing approaches.

**Scalability**. Updating expert mapping incurs non-negligible overhead. Expert replication introduces substantial additional memory consumption, while expert migration incurs extra communication and synchronization costs. These overheads scale with both model sizes and distribution scales. Therefore, the scalability of such approaches are increasingly constrained in large-scale settings.

These limitations motivate the need for a fundamentally different load-balancing strategy—one that does not require modification of the expert mapping and remains both flexible and scalable across diverse inference scenarios.

## 4. Method

To address the aforementioned limitations, we propose Easy-Balance, a cross-layer load balancing strategy for distributed MoE inference. EasyBalance requires no modification to expert mapping, thereby offering superior flexibility and scalability with essentially no additional overheads.

### 4.1. The Cross-Layer Perspective

As discussed in Section 3, the extra memory cost of existing methods primarily stems from expert redundancy. From an intra-layer perspective, redundancy inevitably requires expert replication, which increases memory consumption.

---

**Algorithm 1** EasyBalance

**Input:** Micro-batches of tokens $T_1, \cdots, T_N$, minimum combination size $m$.
\# Record the current layer of each micro-batch.
$C \leftarrow (0, 0, \cdots, 0)$
\# Initialize the variables.
$\mathcal{S}_0 \leftarrow \{T_1, \cdots, T_N\}$
$\text{RunSet}_0 \leftarrow \{T_1, \cdots, T_N\}$
$i \leftarrow 0$
\# The loop continues until all batches finish.
**while** $|\mathcal{S}_i| \neq 0$ **do**
  \# Increase the step.
  $i \leftarrow i + 1$
  \# Run other operations of last-step executed batches.
  *Attn_Route_Dispatch*($\text{RunSet}_{i-1}$)
  \# Choose a $m$-minimum combination.
  $\text{RunSet}_i \leftarrow Schedule(\mathcal{S}_{i-1}, m)$
  \# Execute the scheduled workload.
  *Expert_Compute_Combine*($\text{RunSet}_i$)
  \# Increase $C_j$ of scheduled batches.
  **for** $T_j \in \text{RunSet}_i$ **do**
    $C_j \leftarrow C_j + 1$
  **end for**
  \# Exclude finished batches.
  $\mathcal{S}_i \leftarrow \{T_j | T_j \in \mathcal{S}_{i-1}, C_j < L_{max}\}$
**end while**

---

However, from a cross-layer perspective, we observe that redundancy already exists inherently in the system: experts from other layers can be naturally regarded as "redundant" for the current layer. These experts are already resident in GPU memory, and ready for immediate usage without incurring any additional overhead, which suggests a potential avenue for addressing the current limitations of load balancing methods.

Such potential has been overlooked previously, mainly due to the constraint of layer-wise sequential execution order of models, which prevents experts from other layers from directly participating in the computation. However, there are no layer-wise data dependencies among multiple micro-batches, making it feasible for them to simultaneously reside at different expert layers. As an illustration in Figure 1, even if all batches are at the same layer at step $i$ ($l_j = l$ for $j \in \{0, 1, 2, 3\}$), the yellow micro-batch can be deferred in this step, and at step $i + 1$ the batches will reside on different layers (with yellow batch remaining at layer $l$ while the others advance to $l + 1$). The workloads of batches can also be concurrently executed within one expert-computing stage.

Such cross-layer workload combination is highly likely to create load-balancing opportunities. According to Equation (2), the effective workload of combination of two work-

loads $w_1$ and $w_2$ will be $\max(w_1 + w_2)$, while we have $\max(w_1 + w_2) \leq \max(w_1) + max(w_2)$, meaning that workload combination is always performance-safe even in the worst case. In all other cases, the combined workload is strictly smaller than that of separate execution, indicating its performance benefit. Notably, as the number of devices increases, the probability that multiple workloads share an identical peak device decreases accordingly, thereby reducing the likelihood of the worst-case scenario. The empirical proportion of worst-case cases is reported in Table 2, and the effectiveness of cross-layer workload combination is further demonstrated in Figure 6.

## 4.2. Scheduling

Based on cross-layer expert redundancy and workload combination, EasyBalance mitigates load imbalance through cross-layer micro-batch scheduling. The end-to-end procedure of EasyBalance is demonstrated in Algorithm 1.

At each expert-computation stage, the scheduler selects a subset of micro-batches (Runset$_i$) out of all unfinished ones ($\mathcal{S}_{i-1}$), and executes the combined expert workloads of these batches. Other batches do not participate in the current execution and remain in $\mathcal{S}_i$, waiting for being combined with future workloads from other layers. Only the executed batches that have not yet completed (i,e., $C_i < L_{max}$) proceed to the next layer, progressing through attention, routing, and all-to-all dispatch operations. This iterative process continues until all batches finish execution at their final layer.

As the workload distributions of future layers are unknown at the current layer, the scheduling algorithm must be greedy. We explore several scheduling strategies and find that selecting the workload combination that maximizes computation-resource utilization yields the best performance (see Section 5.2 for the results). Additionally, to avoid executing overly small subsets—where computation and communication overlap becomes ineffective—we introduce a minimum threshold $m$ on the size of RunSet: the scheduling algorithm is required to return a new Runset with $|\text{Runset}| \geq m$. If $|\mathcal{S}| < m$, all available workloads will be executed regardless of imbalance. We suggest the value of $m$ to be $0.5 \sim 0.75 \times$ of the number of micro-batches, according to empirical evidences in Section 5.4.1.

EasyBalance provides instant flexibility across tasks, as all scheduling decisions are made solely based on the current workload distribution and do not rely on any task-specific routing statistics. Moreover, it incurs negligible additional overhead as no expert replication or migration is involved, which poses no limitation upon scalability. The scheduling algorithm operates only on small-sized routing metadata (namely current workload distributions), and its runtime overhead is lightweight and negligible compared to the over-all model inference (see empirical results in Table 3).

# 5. Experiments

**Models**. We evaluate EasyBalance on 3 representative open-source MoE models: Qwen3-30B-A3B-Instruct-2507 (Yang et al., 2025), Moonlight-16B-A3B-Instruct (Liu et al., 2025a) and Qwen3-235B-A22B-Instruct-2507. Except for results in Section 5.3, we use the standard expert-device mapping where experts are evenly distributed across devices. The number of activated experts per token follows the default value in model configurations.

**Benchmark**. We use LongBench (Bai et al., 2024) as the evaluation benchmark. LongBench is designed for large-scale and long-context inference, covering a diverse set of tasks such as passage comprehension, question answering, information retrieval, summarization, and code generation. This task diversity induces highly heterogeneous routing patterns across layers and inputs, making LongBench well suited for evaluating the flexibility and robustness of load-balancing strategies under realistic inference workloads.

**Configurations**. All experiments are conducted on a single node equipped with 8×A100 GPUs 80GB. Inter-GPU communication is handled by NCCL (NVIDIA Corporation, 2017). Unless otherwise stated, the expert parallelism degree is set to 8. The batch size is set to 16 for each GPU, while the token sequence lengths for smaller (Qwen3-30B and Moonlight-16B) and larger (Qwen3-235B) models are 4K and 512 respectively. The number of micro-batch splits is 4. $m$ is set to 3, as this setting typically achieves optimal performances.

**Metrics**. We evaluate inference speed with end-to-end latency, which directly reflects the empirical performances. Following Section 2.3, we additionally report effective workload in Equation (2) and device utilization Equation (3) as quantitative indicators of load imbalance. For readability, we report the reverse number of GPU utilization $(1-u)$ as "under-utilization" most of the time. All measurements are aggregated over 8 runs to avoid system fluctuation, except specification.

## 5.1. Main Results

Figure 2 presents the results of end-to-end latencies and GPU under-utilization of tested models across various tasks. As shown, EasyBalance consistently improves inference performance across model architectures and task categories, significantly alleviating GPU under-utilization caused by expert load imbalance by more than 40% ($\geq 35\%$ to $\approx 20\%$). Its consistent effectiveness across tasks demonstrates strong stability and flexibility in practice, which is attributed to its design that does not require modifications to expert mapping.

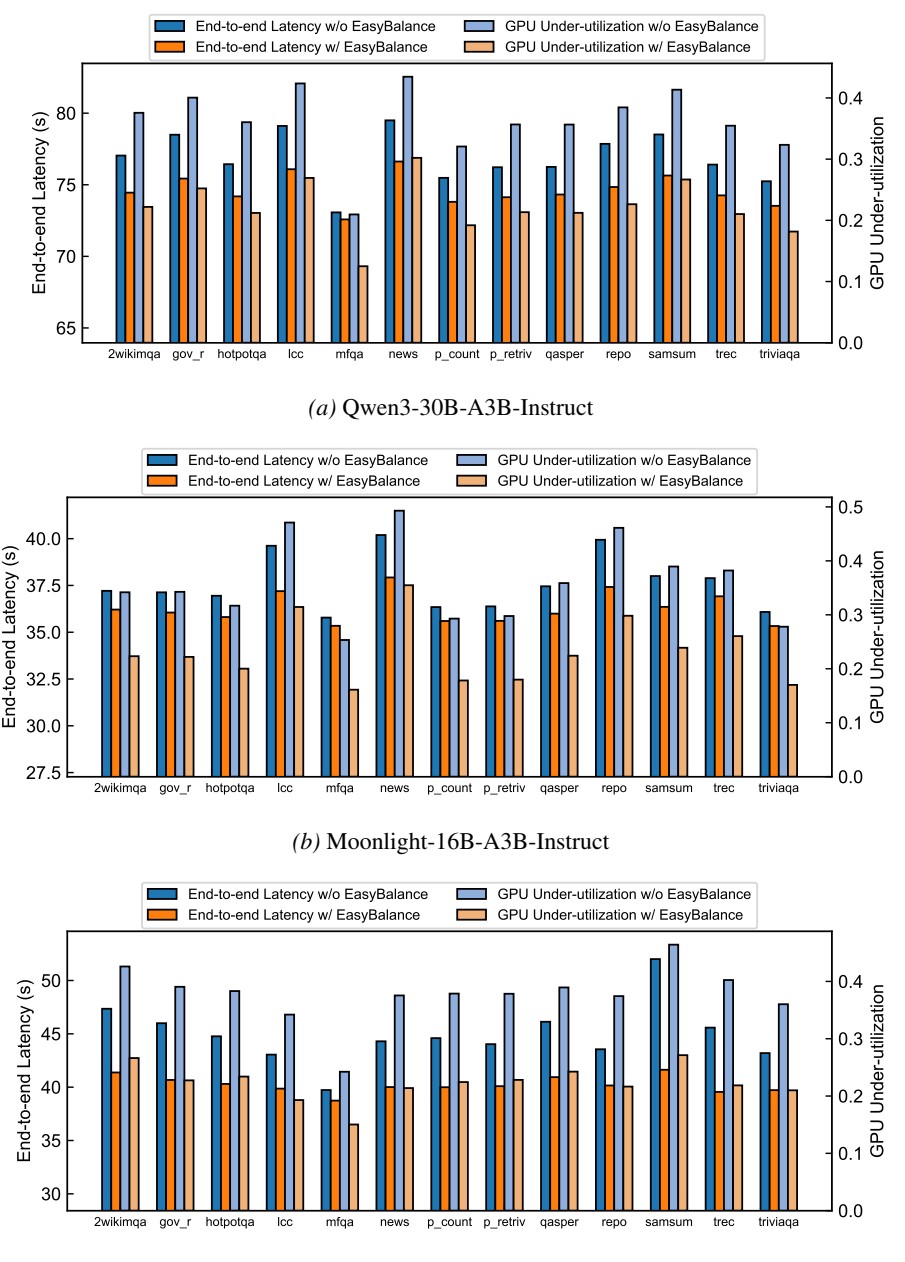

*(a)* Qwen3-30B-A3B-Instruct

*(b)* Moonlight-16B-A3B-Instruct

*(c)* Qwen3-235B-A22B-Instruct

*Figure 2.* End-to-end latency and GPU under-utilization across tasks and models. Lower is better.

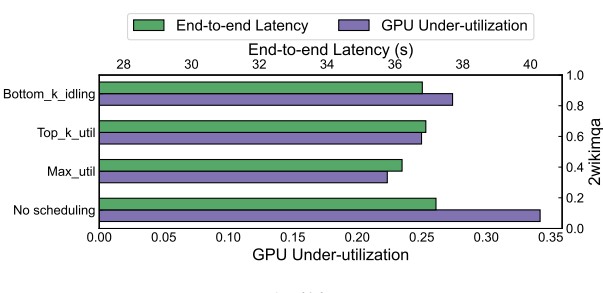

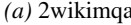

*(a)* 2wikimqa

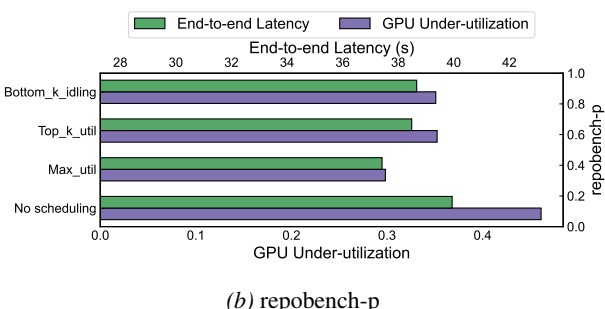

*(b)* repobench-p

*Figure 3.* End-to-end latency and GPU under-utilization of different scheduling algorithms, applied to Moonlight-16B-A3B-Instruct on representative tasks. Lower is better.

| Micro-batches | End-to-end (s) | Scheduling (s) |
|:---:|:---:|:---:|
| 2 | 9.79 | 0.015 |
| 4 | 9.63 | 0.018 |
| 8 | 9.89 | 0.036 |

*Table 3.* Scheduling latencies with max_util strategy of Qwen3-30B on 2widimqa. The batch size and sequence length are 128 and 4K.

## 5.2. Scheduling Algorithms

The scheduling procedure incurs only negligible overhead, as shown in Table 3. However, while the utilization-maximization scheduling strategy achieves strong effectiveness, its overhead grows exponentially with the number of batches. Therefore, we additionally evaluate two alternative heuristic algorithms whose overhead scales linearly.

Denote the workload distribution of micro-batch $i$ across $D$ devices as $w_i = (w_i^0, w_i^1, \cdots, w_i^{D-1})$, the algorithms are:

**Top-$m$ GPU utilization**. Batches are ranked by their individual GPU utilization, and the top-$m$ ones with the highest values are scheduled.

**Bottom-$m$ idling**. Batches are ranked by their individual idling time (measured by $\max(w) - \min(w)$), and the top-$m$ ones with the lowest values are scheduled.

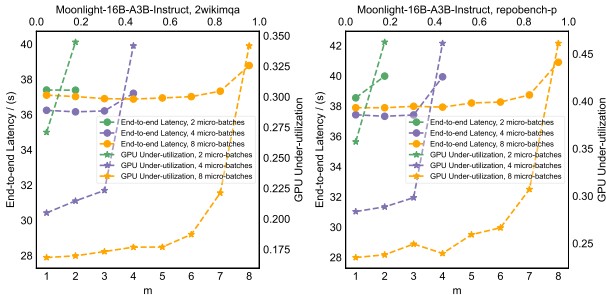

*Figure 4.* End-to-end latency and GPU under-utilization of different micro-batches and $m$ on representative tasks. Lower is better.

Compared to the original algorithm, these two alternatives do not account for the interaction between workload distributions across micro-batches. The results in Figure 3 show that, while these alternatives underperform the maximum-utilization strategy, they both outperform the vanilla baseline (i.e., no cross-layer combination), suggesting that the effectiveness of EasyBalance does not critically depend on a specific scheduling heuristic.

## 5.3. Orthogonality to EPLB

EasyBalance is orthogonal to expert-map modification load balancer, as it does not modify the expert mapping itself. To validate this, we evaluate EasyBalance in combination with EPLB (DeepSeek AI, 2025). We configure EPLB's expert mapping using the routing distribution history from a single task (2wikimqa), without introducing replications. The results in Figure 5 show that EasyBalance consistently provides additive performance gains, confirming their orthogonality.

Figure 6 shows workload statistics of each scheduling step, where the maximum and effective workloads are defined as $\sum_j \max(w_{j,i})$ and $\max \sum_j w_{j,i}$, respectively ($w_{j,i}$ denotes the workload distribution of batch $j$ at step $i$). The results show that EasyBalance effectively mitigates load imbalance through cross-layer workload combination (as stated in Section 4.1).

Importantly, the performance gains of EPLB on similar tasks (e.g., from 2wikimqa to other English NLP tasks) are substantially higher than on other tasks, highlighting the task-dependency and limited flexibility of existing map-modification methods. In contrast, EasyBalance demonstrates consistent and stable effectiveness across all evaluated tasks.

## 5.4. Ablations

The ablation studies are about configurations of micro-batching and parallelism.

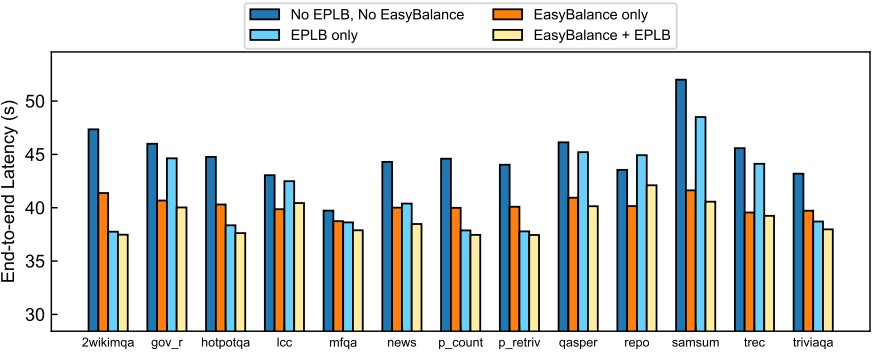

*Figure 5.* End-to-end latency and GPU under-utilization with/without EPLB on Qwen3-235B-A22B-Instruct. The EPLB expert placement is generated from the routing distribution of 2wikimqa, with 128 global experts.

### 5.4.1. MICRO-BATCHING

We vary both the number of micro-batches and the minimum execution threshold $m$. The results, shown in Figure 4, demonstrate that the effectiveness of EasyBalance remains consistent across different micro-batching configurations. The value of $m$ plays a critical role in the scheduling behavior: a smaller $m$ may restrict the scheduler to explore more balancing opportunities, while a larger one may force imbalanced workloads to participate. Based on the empirical results, we suggest the value of $m$ to be $0.5 \sim 0.75\times$ of the number of micro-batches.

Choosing an appropriate micro-batch size is essential for achieving optimal inference performance. Excessively small sizes (e.g., 2) result in insufficient overlap between computation and communication, and may trigger GPU memory throttling, whereas overly large sizes can reduce computation intensity (due to smaller per-batch token budgets) and slow down execution (Leviathan et al., 2023). Notably, optimal performance is achieved with a micro-batch size of 4, while the highest GPU utilization occurs at a size of 8. This suggests the potential that even greater performance gains may be realized under heavier inference workloads (with higher computation intensity).

### 5.4.2. PARALLELISM CONFIGURATIONS

Table 4 reports the value of effective workloads under various expert-parallel (EP) configurations. To ensure a fair comparison across settings, we proportionally adjust the global batch size so that the per-device capacity remains constant (e.g., 32, 64, 128 for EP size = 2, 4, 8 respectively). The results indicate that EasyBalance consistently reduces the effective workloads across all expert-parallel configurations, demonstrating that its benefits are not tied to a specific degree of parallelism.

Notably, configurations with smaller expert-parallel sizes uniformly exhibit lower effective workloads than those with

| Task | Method | Expert-Parallel Size | | |
|---|---|---|---|---|
| | | 2 | 4 | 8 |
| Qwen3-30B-A3B-Instruct | | | | |
| 2wikimqa | w/o EB | 2.23 | 2.56 | 3.22 |
| | w/ EB | 2.10 | 2.28 | 2.59 |
| trec | w/o EB | 2.20 | 2.50 | 3.12 |
| | w/ EB | 2.11 | 2.23 | 2.55 |
| repobench-p | w/o EB | 2.24 | 2.65 | 3.27 |
| | w/ EB | 2.09 | 2.25 | 2.60 |

*Table 4.* Effective workloads (/B tokens) of Qwen3-30B on representative tasks. The batch size for EP=2,4,8 is 32,64,128 respectively. EB stands for EasyBalance.

larger EP sizes,. This behavior arises because expert load imbalance is naturally smoothed when more experts are co-located on a single device. Specifically, a smaller EP size implies that each device hosts more experts, allowing skewed routing distribution to be absorbed locally before manifesting as inter-device imbalance. This observation is consistent with prior studies (OpenLM.ai, 2025), that expert load imbalance becomes increasingly pronounced as expert parallelism scales out. Consequently, load-balancing methods with strong scalability are of greater practical importance in large-scale expert-parallel inference.

## 6. Related Works

**Distributed MoE inference**. Existing works focus on various aspects of distributed MoE inference acceleration. Sc-MoE (Cai et al.) adopts a shortcut connection to break the conventional dependency between communication and computation in distributed MoE models. Tutel (Hwang et al., 2023) designs an identical layout for distributing MoE model for switchable parallelism and dynamic pipelining methods. KTransformers (Chen et al., 2025) developed faster CPU kernels and proposed expert deferral for device-

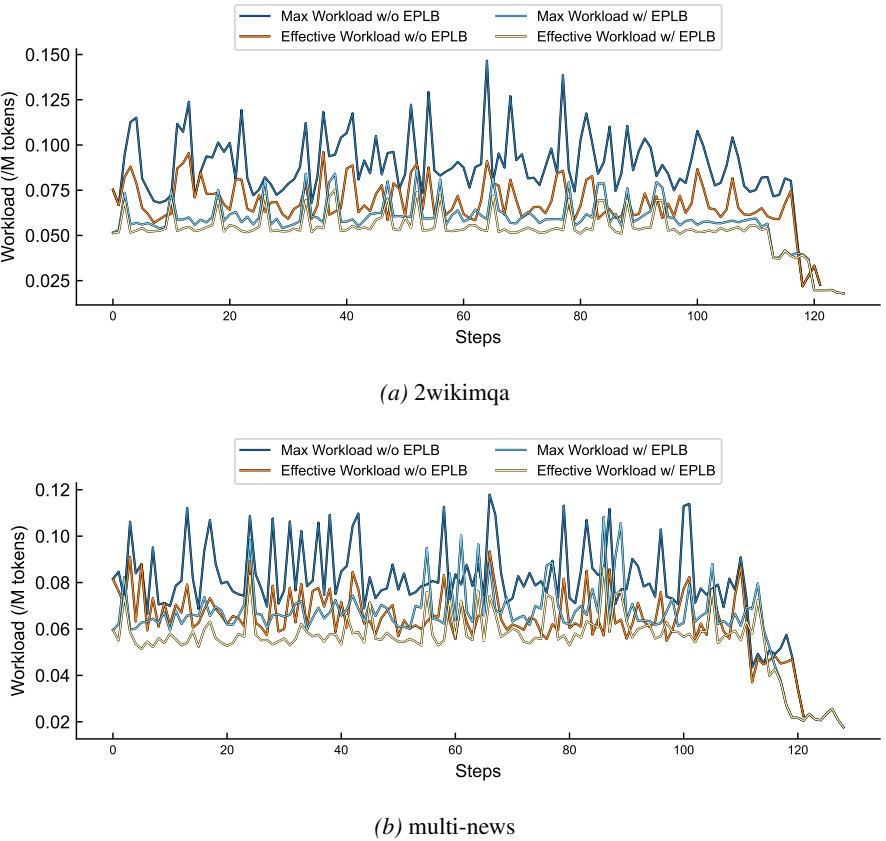

*(a)* 2wikimqa

*(b)* multi-news

*Figure 6.* Maximum and effective workloads of Qwen3-235B on representative tasks in 1 run. The EPLB expert placement is generated from the routing distribution of 2wikimqa, with 128 global experts.

heterogeneous distributed MoE inference. DeepSpeed-MoE (Rajbhandari et al., 2022) implements hierarchical all-to-all based on tensor and expert-parallel topology to reduce communication overheads. Occult (Luo et al.) combine the co-activated experts onto the same device for efficient all-to-all communication.

**Load Balancing**. Works about load balancing are mainly solving it with routing statistic analysis and expert replication or migration. Lina (Li et al., 2023) observed that the token that are routed to the same expert are more likely to be routed to another same expert in the next layer, and use this property to profile routing pattern and guide expert replication. EPLB (DeepSeek AI, 2025) creates expert redundancy and position it with replicated experts, and changes the expert map according to routing histories. Harmoeny (Doucet et al., 2025) reallocates tokens and experts based on current routing results, and migrating experts in the background of computation.

The flexibility and scalability of existing load-balancing strategies are largely limited by expert-device mapping modification, yet no previous solution has been proposed. To the best of our knowledge, we are the first to focus on non-

expert-map modification strategy for load balancing in this area.

## 7. Conclusion

This paper proposes EasyBalance, a cross-layer load balancing strategy for expert-parallel MoE inference. Our key insights involves two observations: cross-layer expert redundancy and workload combination. Instead of relying on expert replication and migration, EasyBalance explores load-balancing opportunities by scheduling different micro-batches to execute on multiple expert layers. Experiments show that EasyBalance can constantly accelerate MoE inference across various inference scenarios, by boosting resource utilization by over mostly 40%.

## Impact Statement

This paper presents work whose goal is to advance the field of Machine Learning. There are many potential societal consequences of our work, none which we feel must be specifically highlighted here.

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

## A. Workload Distribution across Tasks

Figure 7 demonstrates the routing pattern of each layer of Qwen3-30B models, with lighter colors indicating lower workloads. According to the results, the routing distributions are highly task-dependent, posing limitations upon flexibility of load balancing methods that require expert-map modification.

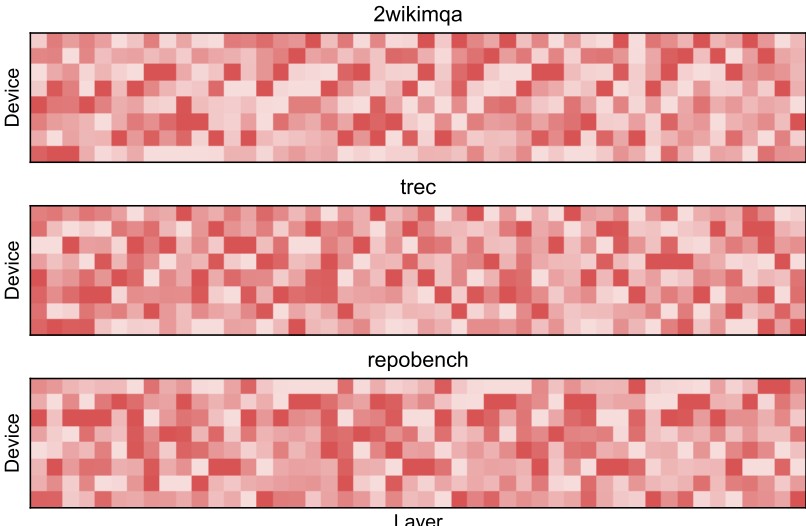

*Figure 7.* Workload distributions across 8 devices of Qwen3-30B-A3B-Instruct on representative tasks. Lighter colors indicate lower workloads.

## B. Clarification of Figures

We report end-to-end latencies as the performance metric, yet EasyBalance targets only at the MoE computation stage, rather than attention. Formally, we have

$$t_{e2e} = t_{expert} + t_{attention}$$

Note that under the same settings of model and (batch size, sequence length), $t_{attention}$ is constant across different tasks (irrelevant to MoE imbalance). Therefore, we set the start of y-axis of figures to be $t_{attention}$ for better demonstration of 'expert latencies'.

## C. More Results

### C.1. Multi-Node Distributed Inference

The results of Qwen3-235B on 2 nodes (8×A100 80GB each) are demonstrated in Table 5, where EasyBalance consistently outperforms the baselines.

### C.2. EPLB with Redundant Experts

EPLB supports redundant expert placement, where redundant experts are replicated from other EP ranks. However, the results in Table 7 show that introducing redundant experts yields only marginal improvements in load balancing, while incurring substantial memory overhead. We evaluate Qwen3-235B-A22B-Instruct with 128 logical experts and observe that adding only two redundant experts already leads to out-of-memory (OOM) issues (144 physical experts for EP=8), while for 136 physical experts the improvement is marginal, especially compared to EasyBalance.

| Task | Method | End-to-end Latency(s) | Effective Workload(M tokens) |
|------|--------|----------------------|------------------------------|
| 2wikimqa | w/o EB | 118.08 | 13.79 |
|          | w/ EB  | 102.93 | 8.33 |
| repobench-p | w/o EB | 111.42 | 12.43 |
|             | w/ EB  | 98.25  | 7.84 |
| trec | w/o EB | 112.26 | 13.27 |
|      | w/ EB  | 100.26 | 8.58 |

*Table 5.* End-to-end latency and effective workloads of Qwen3-235B on representative tasks under 2 nodes. The batch size and sequence length are 32 and 2K for each GPU.

| Task | Method | Expert-Parallel Size | | |
|------|--------|------|------|------|
|      |        | 2 | 4 | 8 |
| 2wikimqa | w/o EB | 0.87 | 1.00 | 1.24 |
|          | w/ EB  | 0.84 | 0.92 | 1.05 |
| trec | w/o EB | 0.88 | 1.03 | 1.32 |
|      | w/ EB  | 0.85 | 0.94 | 1.11 |
| repobench-p | w/o EB | 0.93 | 1.15 | 1.52 |
|             | w/ EB  | 0.86 | 0.97 | 1.17 |

*Table 6.* Effective workload (/B tokens) of Moonlight-16B on representative tasks. The batch size for EP=2,4,8 is 32,64,128 respectively. EB stands for EasyBalance.

### C.3. Scheduling Algorithms

Figure 8 shows more results about different scheduling algorithms. The results indicate that all the scheduling algorithms can reduce end-to-end latency and GPU under-utilization across models and tasks.

### C.4. Workloads

Figure 9 shows results of per-step workload statistics upon Qwen3-235B across more tasks, as complementary to Figure 6.

## D. Limitations

EasyBalance is effective under distributed MoE inference , while it has no acceleration on non-MoE models or single-GPU scenarios (where load imbalance does not exist). EasyBalance currently requires micro-batching, while the potential of applying it to a single sequence (by splitting the sequence into pieces and run them on different layers) remain unexplored. The experiments are conducted only on A100 80GB GPUs, leaving its empirical effectiveness on other platforms uncertain, although the theoretical analysis of effective workload provides strong evidence of its general applicability.

| Task | EPLB(128) | EPLB(128)+EasyBalance | EPLB(136) | EPLB(136)+EasyBalance |
|------|-----------|----------------------|-----------|----------------------|
| 2wikimqa | 2.22 | 1.67 | 2.16 | 1.66 |
| repobench-p | 2.06 | 1.60 | 2.03 | 1.57 |
| trec | 2.20 | 1.69 | 2.21 | 1.60 |

*Table 7.* Effective workload(/M tokens) of EPLB with various numbers of physical experts. The batch size and sequence length are 32 and 512, respectively.

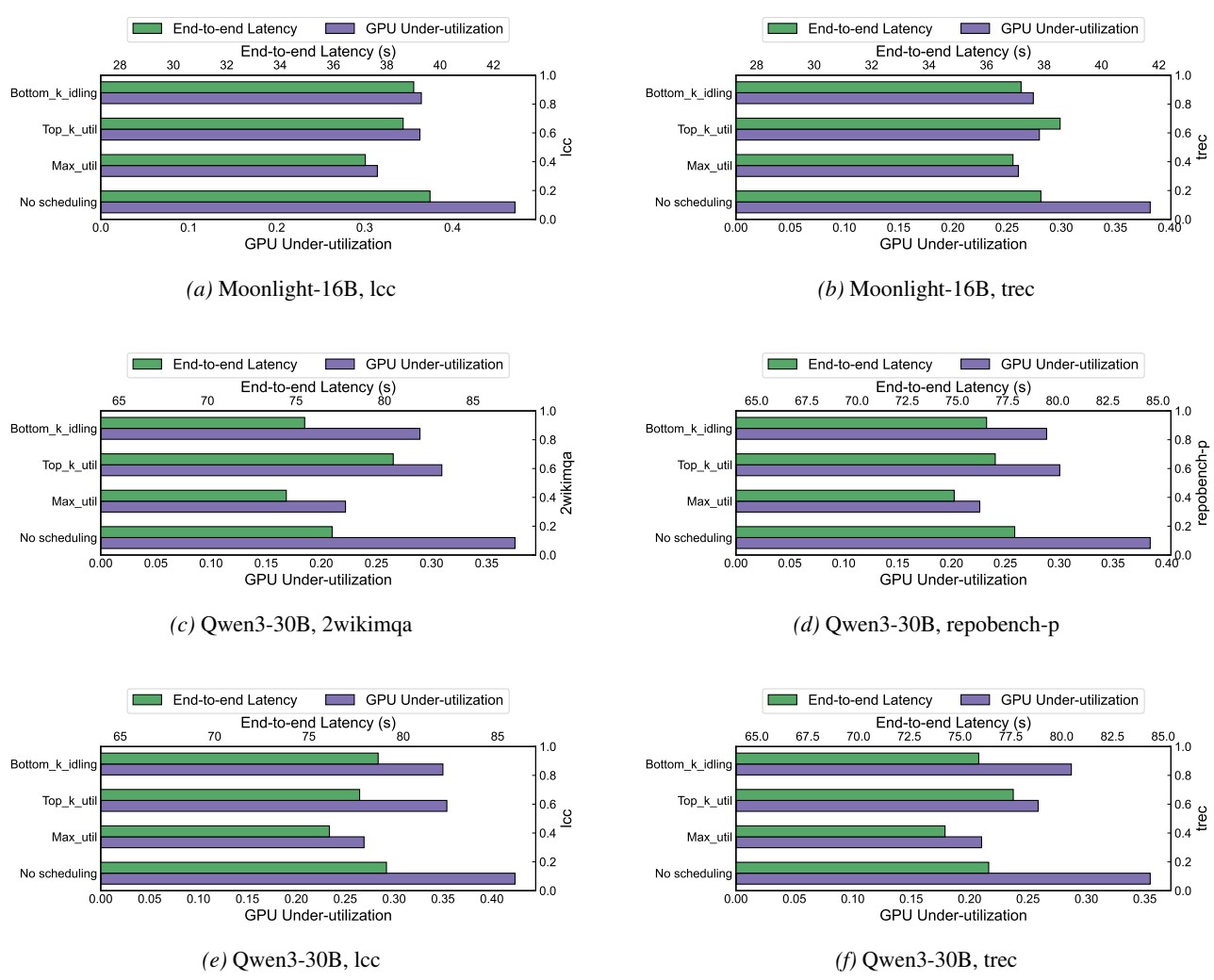

*(a)* Moonlight-16B, lcc

*(b)* Moonlight-16B, trec

*(c)* Qwen3-30B, 2wikimqa

*(d)* Qwen3-30B, repobench-p

*(e)* Qwen3-30B, lcc

*(f)* Qwen3-30B, trec

*Figure 8.* End-to-end latency and GPU under-utilization of different scheduling algorithms across tasks and models.

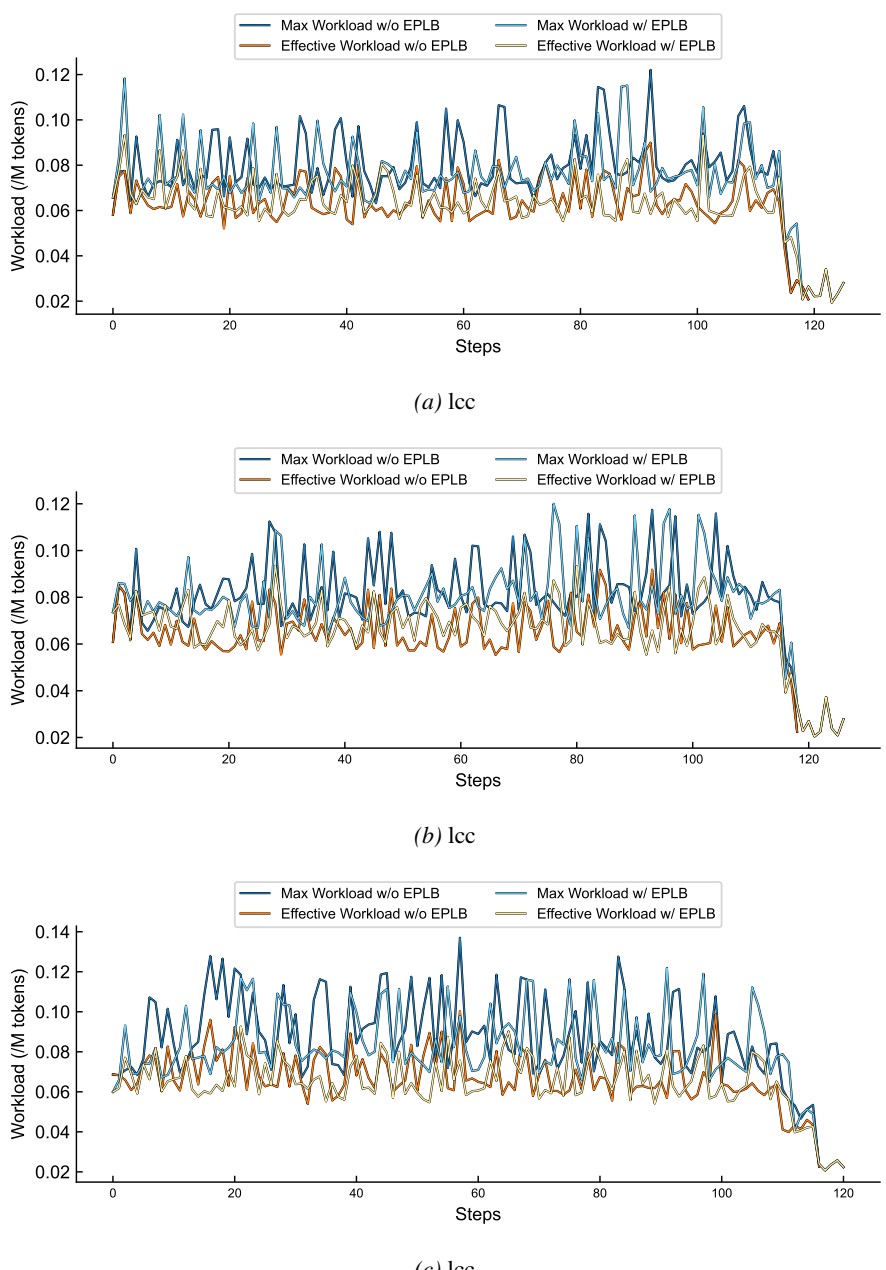

*(a)* lcc

*(b)* lcc

*(c)* lcc

*Figure 9.* Maximum and effective workloads of Qwen3-235B across tasks in 1 run. The expert placement of EPLB is generated from the routing distribution of 2wikimqa, with 128 global experts.

