# OpenReview forum: "EasyBalance: Cross-Layer Load Balancing in Distributed MoE Inference"
_ICML.cc/2026/Conference — ICML 2026 regular_

### Official Review · Reviewer_FGtS · 2026-03-03

**Soundness:** 2
**Presentation:** 3
**Significance:** 2
**Originality:** 2
**Overall Recommendation:** 4
**Confidence:** 5

**Summary:**

This paper proposes EasyBalance, a cross-layer load balancing strategy for expert-parallel MoE inference. The key observation is micro-batches may reside at different expert layers simultaneously, and their expert workloads can be jointly merge and processed. They employ a greedy scheduler to select which micro-batches to execute at each computation stage to maximize GPU utilization. Evaluated on Qwen3-30B-A3B and Moonlight-16B-A3B, EasyBalance reports over 40% reduction in GPU idling.

**Compliance With Llm Reviewing Policy:**

Affirmed.

**Final Justification:**

In the initial review, I was concerned about the lack of end-to-end evaluation, and especially does not consider attention block peformance, which may lead to local improvement but overall worse. Now this concern is addressed.

**Key Questions For Authors:**

1. Delayed micro-batches must later run attention and linear projections alone with a smaller batch, producing memory-bound GEMMs. What is the measured attention layer overhead, and what is the end-to-end speedup?
2. Both models activate only 3B parameters on 8×A100 GPUs. How does EasyBalance perform on larger-scale models (e.g., DeepSeek-V3)?
3. How does EasyBalance compare against existing load-balancing methods such as EPLB (DeepSeek AI, 2025) and Harmoeny (Doucet et al., 2025) cited in the paper? Can it be combined with them for additive gains?

**Limitations:**

Not directly discussed. The impact on attention-layer efficiency due to desynchronized micro-batches should be acknowledged as a limitation.

**Strengths And Weaknesses:**

**Strengths**
1. The cross-layer perspective is genuinely novel. Instead of modifying expert-device mappings via replication or migration, the method exploits an internal property of micro-batched inference.
2. The paper is well-structured. The motivation clearly articulates why existing methods are limited in flexibility and scalability, and the method section logically follows from the key observations.
3. Load imbalance in expert parallelism is a well-known practical bottleneck. A zero-overhead, instantly adaptive solution addresses a real deployment pain point.

**Weaknesses**
1. Delayed micro-batches must later run attention and linear projections alone, producing smaller, memory-bound GEMMs. The overhead is shifted from the expert stage to the attention stage, but only expert latency is reported, this cost is hidden.
2. The heatmap (Figure 2) shows that workload distributions differ by task and layer is already conveyed by the text. The figure occupies substantial space without providing enough information.
3. The experiments use 8×A100 GPUs, yet both models activate only 3B parameters — a mismatch that under-utilizes the hardware. Evaluating on a larger model such as DeepSeek-V3 would be more convincing and better demonstrate the system-level speedup.
4. Without end-to-end latency numbers or comparison against existing load-balancing methods, it is difficult to assess whether the proposed approach yields meaningful practical gains.

---

> ### Author Rebuttal · Authors · 2026-03-31
>
> We sincerely thank you for the review!
>
> > Delayed micro-batches must later run attention and linear projections alone with a smaller batch, producing memory-bound GEMMs. What is the measured attention layer overhead, and what is the end-to-end speedup?
>
> We believe there is a misunderstanding regarding the reported results of '**expert latency**', likely due to an ambiguous definition of it.
>
> In fact, the end-to-end performance improvements (and hence, practical effectiveness of EasyBalance) **have already been demonstrated** with the reported 'expert latency' results. Specifically, with the same model and settings, for two runs (with different scheduling strategies) we have
> $$t_{e2e(1)} - t_{e2e(2)} = t_{expert(1)} - t_{expert(2)},$$
> where $t_{e2e}$ and $t_{expert}$ are end-to-end and expert latency respectively. That is to say, **differences in expert latency are exactly equivalent to differences in end-to-end latency**. Please see the response to Q3 of Reviewer Wh5a for details.
>
> The concern regarding reduced computation intensity is valid, but it is indeed about the micro-batching technology itself, rather than EasyBalance (which only adopts it). In theory, using an extremely large batch size would maximize GEMM FLOPs. However, micro-batching is almost always adopted in practice, due to the need to overlap computation and communication, reduce memory overheads, etc., and therefore it usually produces higher MFU.
>
> We will add these clarifications in the later version.
>
> > Both models activate only 3B parameters on 8×A100 GPUs. How does EasyBalance perform on larger-scale models (e.g., DeepSeek-V3)?
>
> EasyBalance can also be effective on larger-scale models. The results on Qwen3-235B-Instruct and representative tasks are shown below. The test is on 8*A100 with ep=8, bsz=32 and seq=4K.
>
> | Task | EasyBalance | End-to-end Latency(s) | Workload(M token) |
> | :-: | :-: | :-: | :-: |
> | 2wikimqa | no | 34.55 | 1.67 |
> |  | yes | 32.04 | 1.31 |
> | repobench-p | no | 34.75 | 1.54 |
> |  | yes | 31.35 | 1.25 |
> | trec | no | 35.56 | 1.59 |
> |  | yes | 31.21 | 1.23 |
>
> The results show that EasyBalance is effective on large-scale models, both statistically (workload) and empirically (end-to-end latency).
>
> Additionally, the effectiveness on multi-node scenarios is demonstrated in response to Reviewer BnDQ.
>
>
> > How does EasyBalance compare against existing load-balancing methods such as EPLB (DeepSeek AI, 2025) and Harmoeny (Doucet et al., 2025) cited in the paper? Can it be combined with them for additive gains?
>
> EasyBalance is orthogonal to existing expert-map modification balancer, and the benefits can indeed be **additive**. Moreover, the effectiveness of existing balance algorithms is largely task-dependent, while EasyBalance is not.
>
> Below we show the additive performances of EasyBalance with EPLB[1]. We use the routing stats from 2wikimqa as input to EPLB, and test the performances on all three representative tasks. The model is Qwen3-235B-Instruct, with the token budget=32*4K and ep=8.
>
> | Task | EPLB | EasyBalance | End-to-end Latency(s) |
> | :-: | :-: | :-: | :-: |
> | 2wikimqa |  no | no  | 34.55  |
> |  |  no | yes  | 32.04  |
> |  |  yes | no  |  27.74 |
> |  |  yes | yes  | 23.11  |
> | repobench-p |  no | no  | 34.75  |
> |  |  no | yes  | 31.35  |
> |  |  yes | no  | 34.11  |
> |  |  yes | yes  | 29.43  |
> | trec |  no | no  | 35.56  |
> |  |  no | yes  | 31.21  |
> |  |  yes | no  | 33.86  |
> |  |  yes | yes  |  29.86 |
>
> As shown, the expert map generated by EPLB can largely mitigate the imbalance upon the same task, but on different tasks the improvement is limited. EasyBalance outperforms EPLB under cross-task settings, highlighting the necessity and flexibility of EasyBalance that does not rely on expert-map modification.
>
> More importantly, the benefits of the two approaches are complementary, yielding additive performance gains when combined..
>
> > The heatmap (Figure 2) shows that workload distributions differ by task and layer is already conveyed by the text. The figure occupies substantial space without providing enough information.
>
> Thank you for the suggestion! We posted Figure 2 to concretely show empirical evidence of such distribution differences. We will consider put it into appendices in the later version.
>
> [1] Expert Parallelism Load Balancer (EPLB). https://github.com/deepseek-ai/EPLB

---

> > ### Author Rebuttal · Reviewer_FGtS · 2026-04-02
> >
> > Thanks for your rebuttal, I understand other points now. But I still can't understand the first question. Although merging cross-layer experts improves utilization, it reduces the large batch size for attention. Is this still better than running inference with a larger batch directly? Could you tell whether attention becomes slower with multiple smaller batches, or if the overall latency is still dominated by MoE? A clear answer to this would help me reconsider my score.

---

> > > ### Author Response · Authors · 2026-04-02
> > >
> > > Thank you for the feedback, and it is good to know that your concerns have been addressed except for Question 1.
> > >
> > > ***
> > >
> > > > Although merging cross-layer experts improves utilization, it reduces the large batch size for attention. Is this still better than running inference with a larger batch directly?
> > >
> > > We believe that your current understanding is that: micro-batching itself is harmful to performance (worse than a single large batch), while EasyBalance outweighs this harmfulness by better GPU utilization, leading to overall acceleration.
> > >
> > > However, the fact is that: **micro-batching itself can accelerate, while EasyBalance can accelerate even more upon micro-batching.**
> > >
> > > (1) Will micro-batching reduce computation intensity (due to smaller batch size) and be slower than a single batch?
> > >
> > > Yes if memory limit is not reached, and the computation itself will be slower compared to a large batch size.
> > >
> > > (2) Will micro-batching always be harmful to overall performances?
> > >
> > > No. On the contrary, **running with 'appropriate' micro-batching is better for end2end performance than a single batch**, even without EasyBalance. That is because micro-batching can make the communication of each micro-batch **overlapped** with other batches' computation, effectively hiding communication latency---which is often a dominant overhead. As a result, although computation per micro-batch may be slightly less efficient, the overlap with communication typically outweighs this cost, leading to better overall performance.
> > >
> > > This phenomenon has already been demonstrated in Table 1 of the paper, and further verified in all tables below. The resulting speedup is a key reason why micro-batching has become a standard technique in modern inference engines.
> > >
> > >
> > > (3) **Excessively small or large micro-batching size would be harmful**. Overly large size would cause slower execution due to poor computation intensity, while overly small size is also bad due to insufficient overlapping of computation and communication. For example, the best size for '235B, bsz per GPU=8, seq=768' is 2, while for '235B, bsz per GPU=8, seq=1k' it is 4.
> > >
> > >
> > > (4) **Under any micro-batching size, running with EasyBalance is always better than without it**, as the load imbalance is mitigated. EasyBalance can be effectively adopted whenever # micro-batches > 1. It shows superior performances, regardless of whether the batch size itself is optimal.
> > >
> > > In one word, for latency $t$, (1) (2) (3) imply:
> > >
> > > **t(no micro-batching) > t(appropriate micro-batching w/o EasyBalance)**
> > >
> > > and (4) implies:
> > >
> > > **t(any micro-batching w/o EasyBalance) > t(any micro-batching w/ EasyBalance)**
> > >
> > > So, we have:
> > >
> > > **t(no micro-batching) > t(appropriate micro-batching w/o EasyBalance) > t(appropriate micro-batching w/ EasyBalance)**.
> > >
> > > Conclusion: **Choosing the most appropriate micro-batch size can yield the best performance without EasyBalance, while applying EasyBalance on that setting yields an even better performance.** It is also possible that the final best performance is not achieved at the optimal batch size. More splits enable further imbalance mitigation and cound be faster, e.g. 4 vs 2 in Table 1.
> > >
> > > Like we stated, your concern is indeed about micro-batching ((1), (2) and (3)), rather than whether EasyBalance is effective ((4)).
> > >
> > > > Could you tell whether attention becomes slower with multiple smaller batches, or if the overall latency is still dominated by MoE?
> > >
> > > For attention latency, the conclusion is the same as above: appropriate micro-batching is good (which is also better than a large single batch overall), while too large or small micro-batch sizes are bad. Results are in Table 3 and 4 below.
> > >
> > > EasyBalance does not trade off attention latency for MoE acceleration. Instead, it improves MoE computation while keeping attention latency unchanged. This is because the attention computation for each micro-batch is independent and remains unmodified by EasyBalance. Consequently, attention latency is unaffected by whether EasyBalance is applied.
> > >
> > > ***
> > >
> > > The end-to-end latencies of different model sizes, batch sizes and sequence length are show below, with EP=8.
> > >
> > >
> > > Table 1. 235B, bsz per GPU=8, seq=768
> > > | # micro-batches | end-to-end latency(s) |
> > > | :-: | :-: |
> > > | 1 |  5.46  |
> > > | 2 |  4.15  |
> > > | 2+EasyBalance |  **4.11**  |
> > > | 4 |  4.24  |
> > > | 4+EasyBalance |  **4.06**  |
> > > | 8 |  5.54  |
> > > | 8+EasyBalance |  **5.17**  | -->
> > >
> > > Table 2. 235B, bsz per GPU=8, seq=1k
> > > | # micro-batches | end-to-end latency(s) |
> > > | :-: | :-: |
> > > | 1 |  12.97  |
> > > | 2 |  7.40  |
> > > | 2+EasyBalance |  **6.16**  |
> > > | 4 |  6.46  |
> > > | 4+EasyBalance |  **5.50**  |
> > > | 8 |  7.04  |
> > > | 8+EasyBalance |  **6.11**  |
> > >
> > > The attention latencies are below:
> > >
> > > Table 3. 30B, bsz per GPU=8, seq=4k
> > > | # micro-batches | attention latency(s) |
> > > | :-: | :-: |
> > > | 1 |  4.69  |
> > > | 2 |  4.13  |
> > > | 4 |  **4.11**  |
> > > | 8 |  4.30  |
> > >
> > > Table 4. 235B, bsz per GPU=4, seq=2k
> > > | # micro-batches | attention latency(s) |
> > > | :-: | :-: |
> > > | 1 |  4.27  |
> > > | 2 |  3.90  |
> > > | 4 |  **3.88**  |

---

### Official Review · Reviewer_Wh5a · 2026-03-04

**Soundness:** 2
**Presentation:** 2
**Significance:** 3
**Originality:** 3
**Overall Recommendation:** 4
**Confidence:** 4

**Summary:**

This paper tackles the critical load imbalance bottleneck in expert-parallel distributed inference for MoE models, where skewed token routing leads to severe GPU idling, and identifies that existing intra-layer expert replication or migration solutions have inherent flaws in flexibility and scalability with notable extra overhead. It puts forward EasyBalance, a novel cross-layer load balancing strategy built on two core insights: natural redundancy of experts across layers, and the viability of jointly executing expert workloads from different layers of distinct micro-batches. Without any modification to expert-device mapping, EasyBalance adopts greedy cross-layer workload scheduling to alleviate imbalance, realizing instant task adaptability with essentially no additional overhead.

**Compliance With Llm Reviewing Policy:**

Affirmed.

**Final Justification:**

I like this idea, but the execution model in Figure 1 needs more detail to help readers understand key aspects such as how the ActiveSet is obtained and how All-to-All dispatch is overlapped across micro-batches; nonetheless, the rebuttal addressed my main concerns, so I have increased my rating.

**Key Questions For Authors:**

pls see weakness

**Strengths And Weaknesses:**

## Strengths
1. It breaks the traditional intra-layer load balancing paradigm, and its cross-layer design with natural redundancy insight is innovative for me.
2. It achieves lossless inference acceleration with GPU idling reduction.
##  Weaknesses
1. The paper has writing inconsistencies: the batch size annotation (32, 64, 128 for EP size = 1, 2, 4) mismatches Table 3, and the code order on lines 229–230 may need swapping, harming readability.

2. Many critical implementation details are omitted: it does not explain how communication-computation overlap is achieved, how cross-layer GroupGEMM is unified in one operator, or whether extra memory overhead is incurred.

3. The definition of “Expert latency” is unclear, and the paper fails to provide end-to-end (e2e) inference speedup results, making the performance evaluation incomplete and less convincing.

---

> ### Author Rebuttal · Authors · 2026-03-31
>
> We sincerely thank you for the review!
>
> > The paper has writing inconsistencies: the batch size annotation (32, 64, 128 for EP size = 1, 2, 4) mismatches Table 3, and the code order on lines 229-230 may need swapping, harming readability.
>
> Thank you for pointing out the typos.
>
> (1) The batch size setting is 32, 64, 128 for EP size = 2, 4, 8. EP=1 is not involved, as there is no load imbalance without EP.
>
> (2) The code in line 230 should be removed. The newly routed PreExpertSet should be combined with waiting items in the ReadySet, which is then scheduled.
>
> We appreciate your careful reading of the paper and code!
>
>
> > Many critical implementation details are omitted: it does not explain how communication-computation overlap is achieved, how cross-layer GroupGEMM is unified in one operator, or whether extra memory overhead is incurred.
>
> (1) Communication-computation overlap is implemented using two cuda streams, one for computation and one for communication respectively, with cuda events for synchronization.
>
> (2) EasyBalance requires no cross-layer GroupGEMM, as the MoE computation is always within one layer. For each micro-batch, items within it are on the same layer, and cross-layer scheduling is only on micro-batch level.
>
> Besides, the MoE computation of each micro-batch is (and should be) independent with each other, as the result-gathering communication can only begin after its computation completes, which is meanwhile overlapped with the computation of other micro-batches.
>
> (3) No additional memory overhead is incurred compared to the original inference procedure. For each micro-batch, only the output hidden states need to be restored (for the next layer) after each 1-layer forward, regardless of whether other micro-batches reside on the same layer or not. During the internal process of one-layer forward, the memory comsumption is also obviously the same, since the procedures are the same.
>
> We believe these standard implementation details are not the core of our contribution, but engineering specifications. We will put them into appendices in the later version.
>
>
> > The definition of 'Expert latency' is unclear, and the paper fails to provide end-to-end (e2e) inference speedup results, making the performance evaluation incomplete and less convincing.
>
> We believe there is a misunderstanding regarding the reported results of 'expert latency', likely due to an ambiguous definition of it.
>
> In our experiments, we recorded two latency metrics: end-to-end latency ($t_{e2e}$) and attention latency ($t_{attention}$). The attention latency is computed by disable MoE computation (directly 'return' at the first line of MoE function). The reported 'expert latency' is a **derived** quantity, computed as
> $$t_{expert}=t_{e2e} - t_{attention}$$
>
> The tested attention latency ($t_{attention}$) of the models are repoted in the table (under the same settings of ep=8, bsz=128, seq=4K, 8 iterations). Based on these values, the end-to-end latency can be directly obtained by adding the reported expert latency.
>
> | Qwen-30B-A3B | Moonlight-16B-A3B |
> |--- | --- |
> | 64.68s | 28.02s |
>
> Note that $t_{attention}$ is constant under the same settings of model and bsz $\times$ seq (regardless of the task type or scheduling algorithm), because the attention computation workload is independent of expert routing (related only to bsz $\times$ seq). Thus, **differences in expert latency are exactly equivalent to differences in end-to-end latency**:
> $$t_{e2e(1)} - t_{e2e(2)} = t_{expert(1)} - t_{expert(2)},$$
> In conclusion, the end-to-end performance improvements have **already** been demonstrated with the reported expert latency results.
>
> There are some reasons for such settings.
>
> (1) Directly isolating expert computation latency is non-trivial in practice, especially with micro-batching and multi-stream execution.
>
> (2) Attention latency depends on sequence length, whereas MoE computation depends only on the total token budget (bsz $\times$ seq). For instance, workloads of 128$\times$4K budget and 64$\times$8K have the same budget for MoE, but the latter incurs higher attention latency due to the longer sequence length. Therefore, reporting expert-only latency provides more meaningful information about MoE performance.
>
> We will add the clarifications of 'expert latency' in the later version.

---

> > ### Author Rebuttal · Reviewer_Wh5a · 2026-04-05
> >
> > I appreciate the clarification on the latency metrics and the correction of the writing inconsistencies. However, I remain confused regarding the execution model shown in Figure 1.

---

> > > ### Author Response · Authors · 2026-04-06
> > >
> > > Thank you for the feedback, and it is good to know that previous concerns in the review have been addressed.
> > >
> > > > I remain confused regarding the execution model shown in Figure 1.
> > >
> > > For addressing this additional concern, we provide more detailed explanation of both the scheduling procedure and Figure 1.
> > >
> > > ## Scheduling Procedure:
> > >
> > > EasyBalance is a scheduling algorithm, that determines which batches are executed at each step. At any time, each batch belongs to exactly one of the following sets:
> > >
> > > - **ActiveSet**: Batches that are scheduled to perform MoE computation in the next step. ('active' for the next run)
> > >
> > > - **PreExpertSet**: Batches that have completed one-layer MoE step (but not all the layers) and will execute pre-expert operations for the next layer (i.e., attention and routing).
> > >
> > > - **ReadySet**: Batches that have completed attention and routing and are ready for MoE computation schedule. Some of them may not be selected for execution in the current step, and they must wait for further scheduling.
> > >
> > > A batch progresses through the following transitions::
> > >
> > > (1) ActiveSet -> MoE computing -> PreExpertSet (if unfinished)
> > >
> > > (2) PreExpertSet -> attention and routing -> ReadySet (ReadySet = ReadySet $\cup$ PreExpertSet)
> > >
> > > (3) ReadySet -> if scheduled -> ActiveSet; or, if not scheduled, remain in ReadySet (ReadySet = ReadySet - ActiveSet)
> > >
> > > Each batch exits this loop after completing the MoE computation of the final layer $L_{max}$. The overall algorithm terminates once all batches reach $L_{max}$, where the inference completes.
> > >
> > > In summary:
> > >
> > >
> > > **(1) ActiveSet is selected from ReadySet;**
> > >
> > > **(2) ActiveSet performs MoE computation and transitions to PreExpertSet if unfinished;**
> > >
> > > **(3) PreExpertSet executes attention and routing and merges back into ReadySet.**
> > >
> > >
> > > This process repeats until completion.
> > >
> > > ## Explanation of Fig.1:
> > >
> > > (1) The red, purple, blue, and yellow bars represent the workloads of four batches. In EP=4, each workload is partitioned into 4 pieces and executed on 4 devices. Due to non-uniform routing, the per-device workloads are imbalanced (illustrated by longer and shorter lengths of pieces).
> > >
> > > (2) **Without scheduling, all four batches would execute simultaneously**, leading to load imbalance (e.g., Device 2 becomes the bottleneck, while Device 3 is the most underutilized).
> > >
> > > Instead, **EasyBalance selects only three batches to run** (red, blue, and purple), as their combined workloads are relatively balanced. The yellow batch is deferred and placed in ReadySet.
> > >
> > > (ActiveSet: red, blue, purple; ReadySet: yellow)
> > >
> > > (3) The red, blue and purple batch perform MoE computation at layer $l$, then transition to layer $l+1$, entering PreExpertSet to execute attention and routing.
> > >
> > > (PreExpertSet: red, blue, purple; ReadySet: yellow)
> > >
> > > (4) After routing, the updated workload distributions are known, and these three batches are **merged back** into ReadySet (which already contains the yellow batch).
> > >
> > > (ReadySet: red, blue, purple, yellow)
> > >
> > > (5) The scheduler, at this **another scheduling step**, determines that **executing all four batches results in a balanced workload distribution** across devices. Consequently, all four batches are scheduled for execution this time.
> > >
> > > (ActiveSet: red, blue, purple, yellow)
> > >
> > >
> > > ## Additional Notes:
> > >
> > > (1) Fig.1 represents an idealized case where workload imbalance can be perfectly eliminated. In practice, imbalance cannot always be fully eliminated, but EasyBalance significantly mitigates it.
> > >
> > > (2) Batches are not required to remain at the same layer during scheduling. While Figure 1 depicts all batches at layer $l$ initially for simplicity, this is obviously not a constraint.
> > >
> > > (3) Figure 1 focuses on illustration of how load imbalance is mitigated. It omits standard implementation details such as computation-communication overlapping or GroupGEMM.
> > >
> > > Hope these clarifications can be helpful. Thank you!

---

### Official Review · Reviewer_BnDQ · 2026-03-12

**Soundness:** 3
**Presentation:** 3
**Significance:** 3
**Originality:** 2
**Overall Recommendation:** 3
**Confidence:** 3

**Summary:**

This paper targets inference-time load imbalance in expert-parallel Mixture-of-Experts systems, where synchronous gather forces all devices to wait for the most heavily loaded GPU. The proposed method, EasyBalance, does not change expert–device mapping. Instead, it leverages micro-batching to run different micro-batches at different layers concurrently and “combine” cross-layer expert workloads so that device bottlenecks can compensate across layers.

The algorithm maintains a scheduling set of pending expert workloads and greedily selects a cross-layer workload combination that maximizes instantaneous device utilization, subject to a minimum combination threshold. Experiments on two MoE LLMs across LongBench tasks on a single 8×A100 node show reduced GPU under-utilization and lower expert-stage latency, and the paper claims essentially no extra memory or communication overhead.

**Compliance With Llm Reviewing Policy:**

Affirmed.

**Final Justification:**

I do appreciate the authors’ additional results on scheduling overhead, end-to-end latency, multi-node behavior, and sanity checks for extreme routing patterns. However, my main concerns are still not fully resolved: the paper still lacks a direct comparison against a strong MoE inference load-balancing baseline under a clearly common setup, and the novelty relative to related scheduling or co-scheduling approaches remains more argued than demonstrated. Therefore, I am keeping my current score unchanged.

**Key Questions For Authors:**

1. Can authors compare against at least one strong MoE inference load-balancing system, such as replication, migration, or routing-history methods, under the same hardware and model, even if it requires a simplified setup? This is essential for claims of practicality.

2. What is the end-to-end overhead of Schedule() per expert stage, and how does it scale with number of micro-batches and devices? A simple timing breakdown is sufficient.

3. Beyond expert-stage latency and GPU under-utilization, what is the impact on end-to-end request latency and throughput for representative batch sizes?

4. Can authors add one controlled experiment where consecutive-layer skews are highly correlated (peak on the same device) to show worst-case behavior and whether the method degrades gracefully?

**Limitations:**

No. The impact statement is essentially empty. The paper should discuss operational limits: where cross-layer complementarity is absent, how sensitive performance is to routing variability, and whether scheduling could worsen tail latency for certain requests.

**Strengths And Weaknesses:**

Soundness

Strength: The idea is technically plausible and matches the reality of micro-batched inference pipelines where layer progress differs across micro-batches. The “max(w1+w2) ≤ max(w1)+max(w2)” argument gives a clean justification for why combining workloads can reduce straggler effects.

Weakness: The paper does not quantify the scheduler’s own runtime overhead, yet the method is fundamentally an online decision loop. Without measured overhead, it is hard to judge net gains, especially as batch count, layer count, and routing variability scale.

Presentation

Strength: The method is easy to understand at a high level, and the ablations on micro-batching and threshold help interpret behavior.

Weakness: The evaluation protocol is narrow (single-node, limited models), and the paper does not clearly position EasyBalance against the strongest existing MoE inference optimizers beyond the “expert map modification” bucket.

Significance

Strength: Inference-time MoE imbalance is a real bottleneck, and a method that avoids replication or migration is practically appealing.

Weakness: The current evidence does not yet establish that EasyBalance is the right answer in realistic deployments: missing comparisons to established load balancers and missing multi-node results make the claimed deployment relevance premature.

Originality

Strength: Cross-layer scheduling framed as “using redundancy across layers already in memory” is a reasonable systems insight.

Weakness: The “first non-expert-map modification strategy” claim is likely overstated. There is adjacent work on MoE inference scheduling and co-scheduling that appears to address related goals without relying purely on expert relocation. This makes the novelty feel incremental unless the paper clearly differentiates its setting and advantages.

---

> ### Author Rebuttal · Authors · 2026-03-31
>
> We sincerely thank you for the review!
>
> > Q1: Compared to other load balancer
>
> EasyBalance is orthogonal to expert-map changing methods. So instead of comparision, EasyBalance shows **additive** performance gain when applied simutaneously with exisitng load balancers.
>
> Please see the evidence in the response to Reviewer FGtS, who has raised a similar concern.
>
> > Q2: Scheduling overhead
>
> The scheduling time is quite negligible compared to the end-to-end latency. Results on Qwen3-30B with 128*4K budget are shown below:
>
> | Micro-batches | End-to-end | Scheduling |
> | :-: | :-: | :-: |
> | 4 | 9.79s | 0.018s |
> | 8 | 9.83s | 0.034s |
>
> The scheduling time scales linearly with number of devices, and the scaling behavior upon number of micro-batches depends on the algorithm chosen. For 'max_util', it scales exponentially but is the most effective (as shown in Fig4), while for 'top_m_util' and 'bottm_m_idling' it scales linearly and is also effective compared to the vanilla. Empirically, scheduling latency will keep negligible in larger scale, because computation and communication latency also grow with the scale.
>
> > Q3: End-to-end latency
>
> We believe there is a misunderstanding regarding the reported results of '**expert latency**', likely due to an ambiguous definition of it.
>
> In fact, the end-to-end performance improvements (and hence, practical effectiveness of EasyBalance) **have already been demonstrated** with the reported 'expert latency' results. Specifically, with the same model and settings, for two runs (with different scheduling strategies) we have
> $$t_{e2e(1)} - t_{e2e(2)} = t_{expert(1)} - t_{expert(2)},$$
> where $t_{e2e}$ and $t_{expert}$ are end-to-end and expert latency respectively. That is to say, **differences in expert latency are exactly equivalent to differences in end-to-end latency**.
>
> Please see the response to Q3 of Reviewer Wh5a for details.
>
>
> > Q4: Controlled experiments
>
> (1) As we explained in sec4.2. The likelihood of 'consecutive-layer skews peak on the same device' is quite small in real-world scenario. For $N$ devices and $M$ consecutive layers, there are $N^M$ peak-device combinations, with only $N$ among them are 'on the same device'. Therefore, the probability is $1/N^{M-1}$. It is also empirically verified in Table 2.
>
> (2) Even on the condition of 'peak on the same device', the MoE performance will not become worse, only no improvement (as $max(w_1+w_2) \leq max(w_1) + max(w_2)$ in sec4.2). Trivial additional overheads of scheduling do incur, which are yet negligible.
>
> (3) Since 'peak on the same device' rarely occurs in practice, the only way of conducting such controlled experiment would be to artificially **brute-force** expert routing distributions to concentrate on the same device (e.g. by modifying the inference code to enforce this behavior). We argue that such a setup is not meaningful, as it does not reflect any realistic deployment conditions.
>
> In fact, we can even just force the routing distributions to be fully normalized (tokens are evenly distributed across all experts). In this case, load imbalance will be totally 'eliminated', yet the results would be of no practical meaning.
>
>
> > Multi-node results.
>
> EasyBalance remains effective in larger-scale, multi-node settings, both theoretically and empirically. To verify this, we conducted additional experiments on 2 nodes, each with 8*A100 GPUs. The results demonstrate EasyBalance's robust effectiveness. We set bsz=32 and seq=4K. The number of micro-batches and the minimum chunk threshold were set to 8 and 3, following typical configurations.
>
> | Task | EasyBalance | End-to-end Latency(s) | Workload(M token) |
> | :-: | :-: | :-: | :-: |
> | 2wikimqa | no | 118.08 | 13.79 |
> |  | yes | 102.93 | 8.33 |
> | repobench-p | no | 111.42 | 12.43 |
> |  | yes | 98.25 | 7.84 |
> | trec | no | 112.26 | 13.27 |
> |  | yes | 100.26 | 8.58 |
>
> > The 'first non-expert-map modification strategy' claim is likely overstated. There is adjacent work on MoE inference scheduling and co-scheduling that appears to address related goals without relying purely on expert relocation. This makes the novelty feel incremental unless the paper clearly differentiates its setting and advantages.
>
> To the best of our knowledge, our contribution of cross-layer scheduling is novel and unique in this area. There are prior works focusing on request-level imbalance smoothing, which however, all operate in an end-to-end and black-box manner (hoping that imbalance can be smoothed by multiple requests, but without guarantees or algorithmic control). In contrast, our method introduces a deterministic, layer-level scheduling method that explicitly coordinates micro-batches across layers (notably, also requiring no modification of expert map).
>
> We would be happy to provide more detailed comparisons with existing methods, if specific works on which our contribution is possibly considered 'incremental' are identified. We thank the suggestion anyway.

---

> > ### Author Rebuttal · Reviewer_BnDQ · 2026-04-03
> >
> > Thank you for the rebuttal. The added discussion is helpful, especially on scheduling overhead and the extra larger-scale results. However, my main concerns are still not fully resolved. My biggest concern is still the lack of a direct comparison with at least one strong MoE load-balancing baseline under a common setup. This is important for judging the paper’s practical value. I also still think the end-to-end evidence is not yet complete enough. The rebuttal helps clarify the reported latency, but I would still prefer a fuller practical evaluation. In addition, the worst-case behavior is still discussed mainly through argument rather than direct evidence, and the novelty claim would benefit from clearer positioning against related scheduling or co-scheduling work.

---

> > > ### Author Response · Authors · 2026-04-04
> > >
> > > Thank you for the feedback, and it is good to know that some concerns have been solved.
> > >
> > > > My biggest concern is still the lack of a direct comparison with at least one strong MoE load-balancing baseline under a common setup. This is important for judging the paper's practical value.
> > >
> > > In the rebuttal to Reviewer FGtS, we have shown with **empirical results** that EasyBalance is **orthogonal** to exisitng expert-map changing method. We show the **additive** performance of EasyBalance with EPLB[1], which is a strong, typical and commonly used MoE banlancer. As demonstrated, EasyBalance shows **additive** improvements, rather than comparison. For the convenience, we copy the results here:
> > >
> > > Below we show the additive performances of EasyBalance with EPLB. We use the routing stats from 2wikimqa as input to EPLB, and test the performances on all three representative tasks. The model is Qwen3-235B-Instruct, with the token budget=32*4K and ep=8.
> > >
> > > | Task | EPLB | EasyBalance | End-to-end Latency(s) |
> > > | :-: | :-: | :-: | :-: |
> > > | 2wikimqa |  no | no  | 34.55  |
> > > |  |  no | yes  | 32.04  |
> > > |  |  yes | no  |  27.74 |
> > > |  |  yes | yes  | 23.11  |
> > > | repobench-p |  no | no  | 34.75  |
> > > |  |  no | yes  | 31.35  |
> > > |  |  yes | no  | 34.11  |
> > > |  |  yes | yes  | 29.43  |
> > > | trec |  no | no  | 35.56  |
> > > |  |  no | yes  | 31.21  |
> > > |  |  yes | no  | 33.86  |
> > > |  |  yes | yes  |  29.86 |
> > >
> > >
> > > > I also still think the end-to-end evidence is not yet complete enough. The rebuttal helps clarify the reported latency, but I would still prefer a fuller practical evaluation.
> > >
> > > As we have stated, the end-to-end latency of experiments in the paper can be obtained by **simply adding the results to corresponding attention latency**, in response to Reviewer Wh5a. Besides, all the **newly provided** results are indeed end-to-end latency rather than 'expert latency', while EasyBalance shows consistent superior performances.
> > >
> > > To directly show the results, we list the end-to-end latencies of Qwen-30B on tasks below. Note that the numbers can also be directly computed by the expert latency in Fig.3 and the attention latency, as specified in response to Wh5a. The results are obtained from 8 iterations.
> > >
> > > | Task | w/o EasyBalance | w/ EasyBalance |
> > > | :-: | :-: | :-: |
> > > | 2wikimqa | 78.79  | 76.36  |
> > > | gov_report | 80.72  | 77.81  |
> > > | hotpotqa |  78.89 | 76.17  |
> > > | lcc | 82.52 | 78.20  |
> > > | multi_news |  82.61 | 79.27  |
> > > | multifield_qa | 74.34  |  74.02 |
> > > | passage_count | 77.17  | 75.66  |
> > > | passage_retrieval | 78.79  |  76.11 |
> > > | qasper | 78.46  | 76.43  |
> > > | repobench-p | 80.47  | 77.01  |
> > > | samsum | 82.11  | 77.87  |
> > > | trec | 79.12  | 75.99  |
> > > | triviaqa |  76.52 | 75.37  |
> > >
> > > The attention latency is 64.68s, as shown in response to Wh5a. Substracting the results in the above table by 64.68 will get the value in Fig.3 as 'expert latency'.
> > >
> > > > In addition, the worst-case behavior is still discussed mainly through argument rather than direct evidence.
> > >
> > > As said, such worst-case does not exist in real scenario. The only way to do such experiments is to **brute-force the routing results (directly setting its value in the code)**. We have to hardcode it as:
> > >
> > > \# routing results = routing(hidden states) // commented out
> > >
> > > routing results = torch.randint(shape) // brute-force setting the value
> > >
> > > To show concrete evidence, we tested the end-to-end latencies(s) under 2 brute-forcing scenarios:
> > >
> > > (1) normalized (evenly distributed to each expert)
> > >
> > > (2) peaking on device 0
> > >
> > > | Scenario| vanilla | EPLB | EasyBalance |
> > > | :-: | :-: | :-: | :-: |
> > > | normalized | 8.54 |  8.54  | 8.55 |
> > > | peaking | 12.75 |  12.75  | 12.77 |
> > >
> > > The results show that both EPLB and EasyBalance (and basically other balancing methods) would be ineffective under this extreme case, as there is no imbalance (normalized) or the imbalance is unsolvable (peaking). Note that the results are of limited practical meaning, as such scenario never exists in the real world (theoretically explained in sec4.2, empirically shown in Table 2).
> > >
> > > > The novelty claim would benefit from clearer positioning against related scheduling or co-scheduling work.
> > >
> > > We believe that our work is novel against existing 'scheduling or co-scheduling' works.
> > >
> > > What is old: **scheduling batches or requests**. It is generally a common technique, applicable to many scenarios including load balancing.
> > >
> > > What is new: **Layer-level scheduling**. Existing works only schedule the batches as inputs. They decide which requests to be input to the model before each inference step starts, while during the model-forwarding procedure there is no scheduling. Therefore, cross-layer scheduling is indeed novel as our contribution. The observation that **'experts from other layers can be viewed as naturally redundant'** and **'expert workloads of multiple layers from different micro-batches can be jointly executed'** are also new in this area.
> > >
> > > [1] Expert Parallelism Load Balancer (EPLB). https://github.com/deepseek-ai/EPLB

---

### Decision · Program_Chairs · 2026-04-30

**Decision:**

Accept (regular)

**Comment:**

This paper proposes EasyBalance, a cross-layer load-balancing strategy for expert-parallel MoE inference that exploits micro-batching to combine expert workloads from different layers and reduce device-level stragglers. Reviewers found the core idea interesting and practically motivated, and the cross-layer perspective was viewed as the main novelty of the work. The paper also addresses an important bottleneck in MoE serving without requiring expert replication or migration.

The main concerns were about evaluation completeness and positioning relative to prior MoE load-balancing systems. In particular, reviewers asked for clearer end-to-end latency reporting, scheduling overhead measurements, larger-scale / multi-node evidence, and stronger comparison to existing load-balancing approaches. The rebuttal addressed many of these concerns by clarifying the latency metric, providing scheduling overhead numbers, adding larger-model and multi-node results, and showing additive gains when combined with EPLB. This substantially strengthened the practical case for the method.

There are still limitations. The novelty claim should be stated more carefully, and the final paper should better position EasyBalance relative to related scheduling / co-scheduling and expert-map modification methods. Some implementation details and presentation issues also need cleanup. Still, given the positive reception from two reviewers, the resolved concerns after rebuttal, and the practical relevance of the systems contribution, it is a good contribution.